# AlignCLIP: Align Multi Domains of Texts Input for CLIP models with Object-IoU Loss

## ABSTRACT

Since the release of the CLIP model by OpenAI, it has received widespread attention. However, categories in the real world often exhibit a long-tail distribution, and existing CLIP models struggle to effectively recognize rare, tail-end classes, such as an endangered African bird. An intuitive idea is to generate visual descriptions for these tail-end classes and use descriptions to create category prototypes for classification. However, experiments reveal that visual descriptions, image captions, and test prompt templates belong to three distinct domains, leading to distribution shifts. In this paper, we propose the use of caption object parsing to identify the objects set contained within captions. During training, the object sets is used to generate visual descriptions and test prompts, aligning these three domains and enabling the text encoder to generate category prototypes based on visual descriptions. Thanks to the acquired object sets, our approach can construct many-to-many relationships at a lower cost and derive soft labels, addressing the noise issues associated with traditional one-to-one matching. Extensive experimental results demonstrate that our method significantly surpasses the CLIP baseline and exceeds existing methods, achieving a new state-of-the-art (SOTA).

## CCS CONCEPTS

• **Computing methodologies → Matching**.

## KEYWORDS

Multimodal alignment; CLIP; Long-tail learning; Out-of-distribution learning

## 1 INTRODUCTION

Since the inception of Contrastive Language-Image Pre-training [23] (CLIP) by OpenAI, the field of large-scale vision-language pre-training (VLP) has seen rapid advancements. A multitude of approaches [7, 9, 10, 17, 32] have been proposed, achieving remarkable success across a variety of downstream tasks, thereby revolutionizing our understanding and capabilities in bridging visual and linguistic modalities.

Generally, CLIP models are trained using a contrastive learning approach, where they learn to match images with their corresponding text descriptions across a large dataset. This is achieved by

**Unpublished working draft. Not for distribution.**

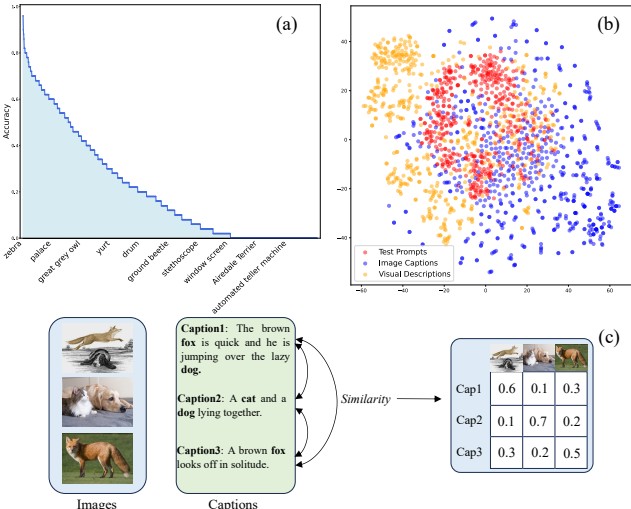

**Figure 1: (a): Zero-shot category-by-category accuracy on ImageNet 1k with pre-training on CC3M. We can see a clear long-tailed distribution with lower accuracy for rare classes in the tail. (b): The distribution of test prompts, image captions and visual descriptions, with obvious distribution shifts among them. (c): Object-IoU loss utilize the IoU of objects in captions to calculate the soft labels. Compared with the use of object detectors in previous methods, the cost is greatly reduced.**

embedding both images and texts into a common space and optimizing the model to bring the embeddings of matching pairs closer, while pushing non-matching pairs apart. During the testing phase, models are usually tested on a variety of tasks such as zero-shot classification, retrieval, linear probe, etc. For the zero-shot classification task, test prompt templates such as "an image of class name" are used to generate prototypes for all categories, based on which images are classified without direct training on those specific categories.

Although the CLIP model has made great progress in recent times, we observe that they are still sub-optimal:

First of all, the distribution of concepts worldwide follows a long-tail distribution. The CLIP models have better zero-shot performance for common concepts and categories. However, their recognition capabilities are significantly weaker for categories that are rare or absent in the training set as shown in Fig.1(a). Using CC3M as pre-training, its recognition performance for many tail categories on ImageNet 1K is very low, and even 295 categories have an accuracy of 0. These categories with an accuracy of 0 are because these categories have never appeared in the training set. For example, the class "Airedale Terrier" in ImageNet 1K have a

accuracy of 0, which is the largest of the terrier breeds, i.e. a kind of dog. However, this fine-grained dog does not appear in the training set CC3M, resulting in its unrecognizability.

A simple and intuitive idea is to use LLM to generate visual descriptions for categories that do not appear in the training set, and use these visual descriptions to generate category prototypes [19, 29], i.e., classifier weights. However, we found that the visual description and the caption of the training set are in two different domains, that is, they have a distribution shift, as shown in Fig.1(b).

Furthermore, from Fig.1(b), we can see the test prompt templates "an image of class name" also distributed differently from training captions. A classic solution is prompt learning [32, 33], that is, learning a prompt template to reduce this domain gap. This can be seen as making the prompts in the test phase align as much as possible with the captions in the training phase. This will cause additional testing overhead and is different from the traditional alignment of the training phase to the testing phase.

In this paper, we solve the above problem in a unified and novel framework, named **AlignCLIP**, that is, we align the image captions, test prompt templates and visual description in the training stage, which enables higher performance during testing with one of them as text input with zero additional overhead. To be specific, during the training phase, we first mining the object items contained in image caption with caption object parsing. The caption object parsing operation takes a caption as input and output the object items set in it. For example, the caption $T_i$ = "*The brown fox is quick and he is jumping over the lazy dog*", which contains two objects, namely *fox* and *dog*, so caption object parsing outputs the set $\pi_i = \{fox, dog\}$. Then, we input the object set $\pi_i$ into two generation modules, namely the visual description generation and the test prompts generation. Specifically, on one hand, we generate visual descriptions for the tail concepts in the training set, so that the text encoder can align the visual descriptions with image features and have the ability to generate category prototypes based on the visual descriptions. On the other hand, we randomly sample templates from the test prompt templates pool to generate test prompts for the set $\pi_i$. Since the set $\pi_i$ may contain more than one object, the features of multiple objects will be averaged and normalized, and then aligned with the image features.

Currently, a popular improvement direction of CLIP is to change the original one-to-one hard label to soft label [9, 10], where different negative samples will be assigned different soft labels according to their similarity, instead of 0 for all. However, existing works all share a common shortcoming, i.e, they use an object detector to detect objects in the image to calculate the similarity, which significantly slows down training, such as SoftCLIP [9]. Thanks to the parsed set $\pi$, we can easily measure the similarity between different negative samples and construct the many-to-many relationship with lower computational cost. According to the object sets in a batch, the intersection over union (IoU) between different sets can be used to measure the similarity between images, and further normalized as soft labels. The loss based on the IoU of parsed object sets is denoted as object-IoU loss.

In summary, our major contributions are as follows:

- We propose to utilize caption object parsing to mine the objects in an image, based on which the tail-classes is enhanced

and different domains of test prompts, image captions and visual descriptions are aligned.
- Based the parsed object sets, the many-to-many relationship is constructed with low cost in object-IoU loss, which makes it easier to generate soft labels and reduce the noise in traditional hard label training.
- We conducted a large number of experiments on multiple open source datasets, and the experimental results proved that our method surpassed existing methods with lower computing consumption.

## 2 RELATED WORK

Here, we show part of the related work. Please refer to the appendix for the complete related work.

### 2.1 Vision-Language Pre-training

In the realm of Vision-Language Pre-training (VLP), the endeavor to synergize visual and textual modalities has been operationalized through extensive training on image-text pairs. Architecturally, VLP models bifurcate into two predominant streams: single-stream and dual-stream frameworks. Single-stream architectures integrate image and text inputs early in the process, utilizing a unified transformer to process the amalgamated embeddings, typified by models such as VisualBERT [14], OSCAR [16], UNITER [5], UNICODER [12], UNIMO [15] and HAMMER [24]. This architecture facilitates direct interaction between modalities within a shared semantic space. Conversely, dual-stream architectures advocate for a modular approach, encoding images and texts through distinct pathways before convergence. Models like CLIP [23], ALIGN [13], DeCLIP [17], SoftCLIP [9], PyramidCLIP [10] and LaCLIP [7] exemplify this approach, underscoring the advantage of discrete yet complementary processing of modal information. Most of this work is to improve certain shortcomings of CLIP. For example, DeCLIP [17] speeds up training through self-supervision. PyramidCLIP [10] uses object detectors for more fine-grained alignment. SoftCLIP [9] uses object detectors to construct many-to-many relationships.

The proposed AlignCLIP belongs to the dual-stream architecture. Differently, AlignCLIP sets out to solve the long-tail distribution in CLIP and misalignment in multiple text domains. Furthermore, we achieve soft label training at low cost based on caption objectf parsing. Compared with previous methods, AlignCLIP training cost is lower and its performance is better.

### 2.2 Long-tail Data Learning

The long-tail distribution [18], where few categories are common and many are rare, presents a significant challenge in data mining and machine learning. Addressing this, researchers have developed three main strategies: re-sampling, re-weighting, and transfer learning. Re-sampling [2, 26, 28] methods adjust the dataset to balance the distribution between common and rare classes, either by increasing the presence of rare classes or reducing that of common ones. Re-weighting [4, 22, 30] approaches alter the loss function to prioritize rare classes during training, giving them more importance. Transfer learning [20, 21, 31] techniques use the knowledge gained from common classes to improve the learning of rare classes, enriching their feature representation. These strategies, from adjusting

data distribution to modifying training emphasis, offer pathways to mitigate the long-tail problem, aiming for a more balanced learning across classes.

However, in multi-modal pre-training, there are relatively few solutions to the long-tail problem, which has been grossly ignored. Although there are some works that use visual descriptions to improve the performance [19, 29], however, they only generate category attributes at the test stage, which leads to the multi-domain misalignment, limiting model performance. We propose to use visual descriptions while solving the distribution shift of multiple domains during the training stage, achieving better results.

## 3 METHOD

In this section, We first introduce the framework of AlignCLIP. Then we introduce the first step of AlignCLIP, which is caption object parsing. After that, we introduce how to perform multi-domain alignment on the basis of parsed object sets. Finally, we introduce how to perform soft label training based on object-IoU loss.

### 3.1 Framework

The training of AlignCLIP is shown in Fig. 2, which is similar to CLIP [23], that is, matching between image and text pairs is trained through contrastive learning. For the convenience of presentation, only a single image and text are shown in Fig. 2 as an example.

To begin with, an image caption $T_i$ is input into the caption object parsing module to obtain the object items in the caption, and the parsed objects set is denoted as $\pi_i$. With set $\pi_i$, on the one hand, we generate the visual descriptions of objects in $\pi_i$, if any of objects in $\pi_i$ belongs to the statistical tail concepts. On the other hand, for each object in the collection $\pi_i$, we randomly sample from the prompt templates pool during testing to generate a simulated test prompt, which is used to solve the distribution shift of training and testing. Since a caption may contain multiple objects, we average obtained multiple visual descriptions and then perform the normalization operation, and the same was done for generated test prompts. Meanwhile, the image $I_i$ is extracted feature with image encoder, which will be aligned with the feature of caption $T_i$, averaged visual description and test prompts at the same time. Finally, we propose object-IoU loss to generate soft labels, which utilize the similarity between object sets in a batch to construct many-to-many relationships.

For model inference, its cost is similar to ordinary CLIP and does not require additional calculations. During inference, the model can support both visual description input and ordinary test prompts. Users can choose which input form to use based on whether the category is common and accuracy metrics.

### 3.2 Caption Object Parsing

In order to perform subsequent multi-domain alignment and object-IoU loss, we first need to perform caption object parsing. Compared with previous methods that perform object detection on input image, obtaining object information from the caption is cheaper and faster, and does not rely on a trained object detector.

Formally, for an image caption $T_i$, the goal of caption object parsing is to obtain all objects $\pi_i = \{o_1^i, o_2^i, ..o_{k_i}^i\}$ contained in the

$T_i$, where $\pi_i$ is the objects set, $o_j^i$ is an object in $\pi_i$ and $k_i$ is the number of items in $\pi_i$.

There are many ways to obtain objects in a sentence. In this paper, we include two methods, namely part-of-speech tagging (POS) and large language model (LLM).

The task of part-of-speech tagging is to mark the part-of-speech of each word in a sentence, such as nouns, verbs, adjectives, adverbs, etc. Usually, the objects in a caption are nouns. Therefore, we use the part-of-speech tagging algorithm in NLTK [1] to get the nouns in the caption and output it as the result $\pi_i$.

For LLM, we design prompts to enable open source LLM to output the objects in the caption. Due to the simplicity of the task, even an extremely small-sized open source LLM can output very high quality results. In the experiments, for the LLM scheme, we use the Gemma-2B-Chat [27] model, and the specific prompts can be found in the appendix.

### 3.3 Multi-Domain Alignment

As mentioned above, the original image caption, the test prompts such as "an image of class name", and the visual description belong to three different domains, and there is a distribution shift between them, which leads to a performance drop. Here, we align these three with image features at the same time during training, so that the model has the ability to use visual descriptions, test prompts, and original captions to generate category prototypes without any distribution shift.

*3.3.1 Tail-class Description Generation.* To align visual descriptions and visual features, a simple approach is to generate visual descriptions for all objects in the training set, and perform the alignment training. However, this consumes a lot of time, since the training set is large. Here, we adopt a more lightweight approach, which is to generate visual descriptions for only the rarest $\alpha$ percent of the tail, where $\alpha$ is a hyperparameter set to 30% in our experiments.

**Object frequency statistics:** In order to distinguish which objects in the training set are rare objects in the tail, our first step is to count the frequency of occurrence of objects in the training set and regard objects with less frequency as rare objects.

To be specific, we first generate a frequency table $\Theta$ that records each object and the number of occurrence the object appears in the training set. Specifically, $\Theta$ is formulated as follows:

$$\Theta = \{(o_j, x_j)\}_{j=1}^{N_1}, \quad \text{where } x_j = \sum_{i=1}^{N_2} \mathbb{I}(o_j \in \pi_i) \quad (1)$$

where $x_j$ is the frequency of occurrence of object $o_j$, $N_1$ is the number of objects in total and $N_2$ is the number of captions in training set. Then, we sort the objects in $\Theta$ according to the number of occurrences $x_j$, and retain the $\alpha$ percentile objects with the least occurrences, which is denoted as $\Theta_\alpha$.

**Description generation:** Since the object set $\pi_i$ of a caption may contain multiple objects, therefore, only a part of the objects in the set $\pi_i$ may belong to the tail objects $\Theta_\alpha$. In this case, our approach is that as long as $\pi_i$ contains any tail object, other objects also generate visual descriptions together. Specifically, for the set

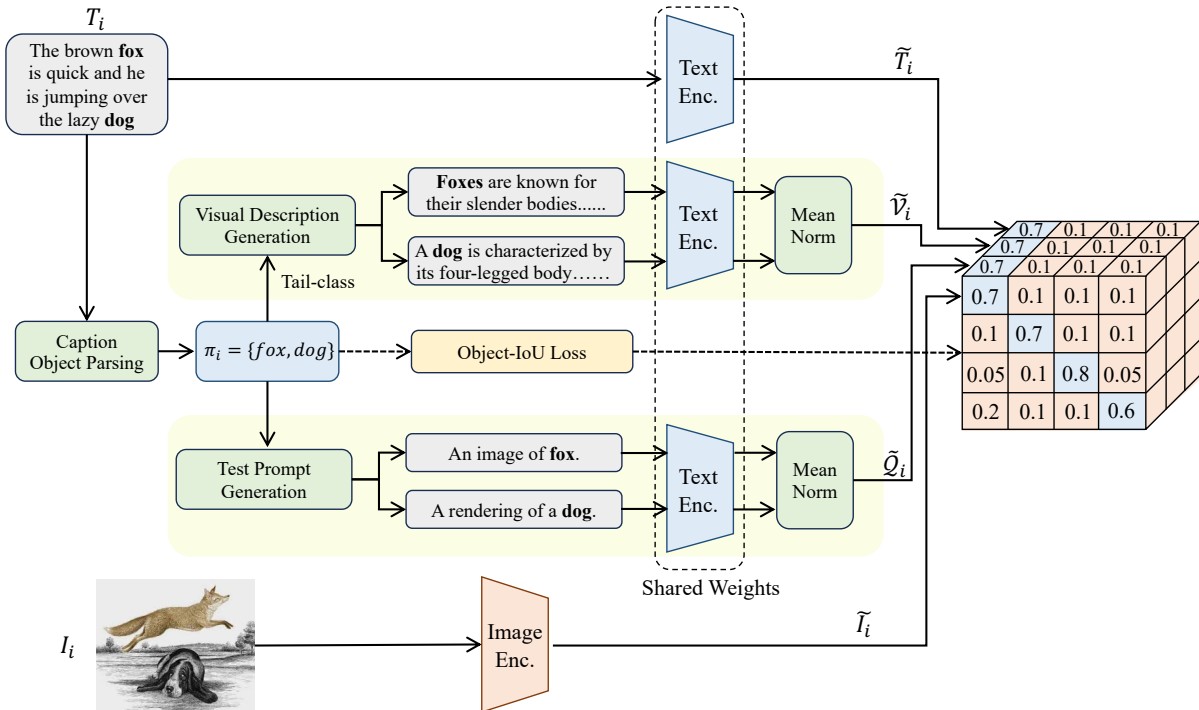

**Figure 2: The training of AlignCLIP. Firstly, for an image caption $T_i$, we perform caption object parsing to obtain the object set $\pi_i$, as $\pi_i = \{dog, fox\}$ above. Based on the set $\pi_i$, we perform multi-domain alignment, that is, align the distribution of original captions, visual descriptions, and test prompts to image features. To achieve this, we need to generate visual descriptions of $\pi_i$ if any of objects in $\pi_i$ belongs to tail-classes, as well as test prompts. Since set $\pi_i$ may contain more than one objects, the mean and normalization operation will be performed to obtain the global feature. Finally, the image feature is trained to align with text features from origin caption, visual descriptions and test prompts under the supervision of object-IoU loss.**

$\pi_i$ of caption $T_i$, its visual description set $\mathcal{V}_i$ is formalized as:

$$\mathcal{V}_i = \begin{cases} \{v_j^i = \mathcal{J}_{vis}(o_j^i) | \forall o_j^i \in \pi_i\}, & \text{if } \exists o_j^i \in \Theta_\alpha \\ \varnothing, & \text{otherwise} \end{cases} \quad (2)$$

where $\mathcal{J}_{vis}$ is a function that inputs an object $o_j^i$ and outputs an appearance description of the object. We implement $\mathcal{J}_{vis}$ with LLM, which is Gemma-2B-Chat [27] and the prompt instruction is in the appendix.

Then, the description features in $\mathcal{V}_i$ will be averaged and normalized to get the global feature that can be aligned with the whole image feature:

$$\widetilde{\mathcal{V}}_i = \begin{cases} \phi\left(\frac{1}{k_i} \sum_{v_j^i \in \mathcal{V}_i} f_t(v_j^i)\right), & \text{if } \exists o_j \in \Theta_\alpha \\ \varnothing, & \text{otherwise} \end{cases} \quad (3)$$

where $\phi(\cdot)$ is the normalization operation. $f_t$ is the text encoder. $k_i$ is the number of objects in set $\pi_i$. Here, we first input the features of multiple descriptions into the text encoder separately, and then average and normalize them to obtain a global descriptions representation of multiple objects. Another simple method is to concatenate multiple visual descriptions and then input to text encoder. However, this method can easily causes text to exceed window length because the visual descriptions are usually very long.

*3.3.2 Test prompt Generation.* As mentioned above, we found that the distribution of test prompts and training captions are in two different domains, and there is a distribution shift between them. Existing work attempts to utilize the prompt learning to learn a better prompt template to splice in front of the category name, such as CoOp [33] and CoCoOp [32]. This learned prompt can better conform to the distribution of captions in the training set, thereby improving performance. However, each test set must be learned separately, which increases the cost of inference.

Here, we follow the idea of data augmentation and generate simulated test prompts based on the parsed object set $\pi_i$, and perform alignment during the training phase. Thus, the test prompts will be aligned with image features, and the domain gap will be narrowed.

Since the zero-shot classification averages the features of multiple prompt templates to obtain the final category prototype in CLIP, we also need to align multiple prompt templates here. We denote the prompt template collection used in model inference as $\mathcal{P} = \{\rho_1, \rho_2, ..., \rho_m\}$, where $m$ is the number of templates. For each object $o_j^i$ in the set $\pi_i$, we randomly sample a template $\rho_h$ from the collection $\mathcal{P}$ to generate a simulated test prompt as follows:

$$Q_i = \{q_j^i = g(\rho_h, o_j^i) \mid \forall o_j^i \in \pi_i, \rho_h \sim \text{Uniform}(\mathcal{P})\} \quad (4)$$

where $g(\cdot, \cdot)$ is a function that combines templates and objects to format a simulated test prompt $q_j^i$. Then, the features of the test prompt $Q_i$ will be extracted by encoder $f_t$, and then averaged and normalized to obtain global features $\widetilde{Q}_i$ as follows:

$$\widetilde{Q}_i = \phi\left(\frac{1}{k_i} \sum_{q_j^i \in Q_i} f_t(q_j^i)\right) \tag{5}$$

In experiments, we take OpenAI 80 templates for ImageNet as $\mathcal{P}$.

## 3.4 Object-IoU Loss

In this section, we first present the CLIP preliminaries, then, we introduce how to build a many-to-many relationship based on the IoU of object sets and perform soft alignment.

*3.4.1 CLIP Preliminaries.* Given a batch of $N_B$ image-text pairs $\{I_i, T_i\}_{i=1}^{N_B}$, CLIP first feeds the image $I_i$ and text $T_i$ and into the image encoder $f_I$ and the text encoder $f_t$ respectively, together with the normalization operation as follows:

$$\widetilde{I}_i = \phi(f_I(I_i)) \tag{6}$$

$$\widetilde{T}_i = \phi(f_t(T_i)) \tag{7}$$

Based on the obtained image embedding and text embedding pairs $\{(\widetilde{I}_i, \widetilde{T}_i)\}_{i=1}^{N_B}$, CLIP uses contrasting learning with InfoNCE for cross-modal alignment, which bring corresponding images and text embedding closer, and pull those that don't correspond farther away.

With embedding pairs $\{(\widetilde{I}_i, \widetilde{T}_i)\}_{i=1}^{N_B}$, the image-to-text and text-to-image similarity matrix can be calculated with:

$$p_{ij}(\widetilde{I}_i, \widetilde{T}_i) = \frac{\exp(sim(\widetilde{I}_i, \widetilde{T}_i)/\tau)}{\sum_{j=1}^{N_B} \exp(sim(\widetilde{I}_i, \widetilde{T}_j)/\tau)} \tag{8}$$

$$p_{ij}(\widetilde{T}_i, \widetilde{I}_i) = \frac{\exp(sim(\widetilde{T}_i, \widetilde{I}_i)/\tau)}{\sum_{j=1}^{N_B} \exp(sim(\widetilde{T}_i, \widetilde{I}_i)/\tau)} \tag{9}$$

where $\tau$ is a learnable temperature parameter that controls the concentration level of the distribution. the function $sim(\cdot)$ measure the similarity of embedding with dot product.

Then, the $i^{th}$ image-to-text similarity vector is denoted as $\mathbf{p}_i(I, T) = \{p_{ij}(I, T)\}_{j=1}^{N_B}$ and the text-to-image similarity vector is denoted as $\mathbf{p}_i(T, I) = \{p_{ij}(T, I)\}_{j=1}^{N_B}$. Then, the hard label are used to calculate InfoNCE loss as follows:

$$\mathcal{L}_{I2T} = \frac{1}{N_B} \sum_{i=1}^{N_B} H(\mathbf{y}_i, \mathbf{p}_i(I, T)) \tag{10}$$

$$\mathcal{L}_{T2I} = \frac{1}{N_B} \sum_{i=1}^{N_B} H(\mathbf{y}_i, \mathbf{p}_i(T, I)) \tag{11}$$

where $\mathbf{y}_i = \{y_{ij}\}_{j=1}^{N_B}$ is one-hot label for $i^{th}$ pair with $y_{ii} = 1$ and $y_{ij(j \neq i)} = 0$. $H(\cdot, \cdot)$ is the cross entropy loss function. The final training loss of the original CLIP is calculated by averaging the Eq.(10) and Eq.(11) as follows:

$$\mathcal{L}_{CLIP} = \frac{1}{2}(\mathcal{L}_{I2T} + \mathcal{L}_{T2I}) \tag{12}$$

*3.4.2 Object-IoU.* As mentioned above, the simple one-to-one assumption does not hold in practice. Different negative samples also have different similarities, and should not simply be assigned a label of 0. PyramidCLIP [10] utilize label smoothing to soften the hard label, which is naive and the improvement is limited. SoftCLIP [9] model the many-to-many relationship with the assistance of object detector, which is onerous and inefficient.

Here, we exploit the parsed object sets $\pi_i$ to cost-effectively construct many-to-many relations. Specifically, for the object set $\{\pi_i\}_{i=1}^{N_B}$ parsed from multiple captions in a batch, the similarity between $\pi_i$ and $\pi_j$ can be measured as:

$$IoU_{ij} = \frac{|\pi_i \cap \pi_j|}{|\pi_i \cup \pi_j|} \tag{13}$$

where $|\cdot|$ represents the size of the collection. Eq.(13) calculates the similarity of two sets through the intersection of union (IoU) of $\pi_i$ and $\pi_j$. Generally speaking, the greater the IoU of two sets, the greater the similarity.

*3.4.3 Soft Alignment.* Based on the target similarity matrix $IoU_{ij}$, we align the image features with the original caption, visual description and test prompt respectively.

For $i^{th}$ object set $\pi_i$, the target similarity vector is denoted as $IoU_i = \{IoU_{ij}\}_{j=1}^{N_B}$. Following previous work [9], we utilize KL-Divergence as loss function. The alignment of image caption $\widetilde{T}_i$, visual description $\widetilde{V}_i$ and test prompt $\widetilde{Q}_i$ with images feature $\widetilde{I}_i$ can be expressed as follows:

$$\mathcal{L}_{IoU-I2\mathcal{T}} = \frac{1}{N_B} \sum_{\mathcal{T} \in \{\widetilde{T}_i, \widetilde{V}_i, \widetilde{Q}_i\}} \mathbb{I}(\mathcal{T} \neq \varnothing) \sum_{i=1}^{N_B} KL(IoU_i || \mathbf{p}_i(\widetilde{I}_i, \mathcal{T})) \tag{14}$$

$$\mathcal{L}_{IoU-\mathcal{T}2I} = \frac{1}{N_B} \sum_{\mathcal{T} \in \{\widetilde{T}_i, \widetilde{V}_i, \widetilde{Q}_i\}} \mathbb{I}(\mathcal{T} \neq \varnothing) \sum_{i=1}^{N_B} KL(IoU_i || \mathbf{p}_i(\mathcal{T}, \widetilde{I}_i)) \tag{15}$$

where $\mathcal{T}$ is variable and $\mathcal{T} \in \{\widetilde{T}_i, \widetilde{V}_i, \widetilde{Q}_i\}$. $\mathbb{I}(\cdot)$ is indicator function, which is used to skip the situation where the visual description is $\varnothing$.

Then, the object-IoU loss of the image and origin caption is the average of the above two formulas:

$$\mathcal{L}_{IoU} = \frac{1}{2}(\mathcal{L}_{IoU-\mathcal{T}2I} + \mathcal{L}_{IoU-I2\mathcal{T}}) \tag{16}$$

Finally, the model is trained by a mixture of original CLIP loss $\mathcal{L}_{CLIP}$ in Eq.(12) and object-IoU loss in Eq.(16) as follows:

$$\mathcal{L}_{AlignCLIP} = \frac{1}{2}(\mathcal{L}_{IoU} + \mathcal{L}_{CLIP}) \tag{17}$$

## 3.5 Model Inference

During model inference, the text encoder can accept test prompts, visual descriptions, and captions at the same time, without any distribution shift.

In the zero-shot classification experiments in this paper, we first perform inference on all categories using the test prompt to generate category prototypes. Then, for classes with lowest $\beta$ percentile, we generate visual descriptions for them and utilize visual descriptions to generate category prototypes, where $\beta$ is a hyperparameter set to 50% in our experiments.

# 4 EXPERIMENTS

## 4.1 Pre-training Datasets and Architectures

Following previous works [9, 10], we have pre-trained on multiple open source mainstream image-text datasets, including Conceptual Captions 3M (CC3M) [25], Conceptual Captions 12M (CC12M) [3], YFCC15M-V2 [17]. For the model architecture, we chose 3 different image encoders, i.e. ResNet-50 [11], ViT-B/32 [6] and ViT-B/16 [6]. Meanwhile, all images are uniformly scaled to 224×224 before being input to the image encoder. For the text encoder, we used the same text encoder with Transformer architecture as CLIP [23], and the maximum token length is set to 77.

## 4.2 Implementation Details

All models are trained in parallel with 8 V100 GPUs. The proportion of tail classes in the training set and test set, i.e. $\alpha$ and $\beta$ is set to 30% and 50%, respectively. We use the AdamW optimizer with the learning rate set to 5e-4. The weight decay rate of AdamW is set to 0.2. Following previous works [9], cosine learning rate scheduler with a linear warm-up is used to adjust learning rate, where the warm-up takes up about 10% of the total steps to increase the learning rate from 0 to the peak value, and then decreases with a cosine anneal strategy. Meanwhile, we use automatic mixed-precision [8] to train the model to save GPU memory. For CC3M and CC12M, we set the batch size to 1024. For YFCC15M-V2, we set the batch size to 2048. For caption object parsing, we implement it with LLM for comparison and detailed ablation studies can be found in experiments. For more details, please refer to our code.

## 4.3 Evaluation Setup

We verified the effectiveness of AlignCLIP on 3 different downstream tasks, namely zero-shot image classification, linear probing and zero-shot image-text retrieval. For zero-shot classification, we utilize the identical prompt templates outlined in the CLIP [23] paper, After the category name is combined with the prompt template, the features will be extracted with the text encoder and used as the category prototype. Images will be classified by calculating similarity with these category prototypes. For linear probing, the image encoder is freeze and a linear layer is added at the end of the image encoder for fine-tuning. For zero-shot image-text retrieval, caption and image features are extracted separately, and the search result with the highest similarity is output.

## 4.4 Zero-shot Image Classification

In order to verify the performance of the proposed AlignCLIP, we first compare the performance of zero-shot classification. Here, the caption object parsing is implemented with Gemma.

We first present the zero-shot classification results on ImageNet 1K in Tab.1. As we can see, the performance of our AlignCLIP has been greatly improved based on the original CLIP [23]. Our AlignCLIP achieves the highest performance across all pre-training datasets and different image encoder settings. Specifically, using ResNet50 as image encoder, AlignCLIP exceeded the original CLIP by 9.2, 11.4, and 7.1 points respectively when pre-trained with CC3M, CC12M, and YFCC15M. Meanwhile, our method also achieves the highest performance using other image encoders. Not only that,

**Table 1: Results of zero-shot classification on ImageNet 1K.**

| Pretrain Dataset | Image Encoder | Method | ImageNet ZS Top1 |
|---|---|---|---|
| CC3M | ResNet50 | CLIP [†] | 17.8 |
| | | SoftCLIP | 24.2 |
| | | **AlignCLIP** | **27.0** |
| | ViT-B/32 | CLIP[†] | 11.8 |
| | | SoftCLIP | 13.3 |
| | | **AlignCLIP** | **15.7** |
| | ViT-B/16 | CLIP[†] | 15.9 |
| | | SoftCLIP | 18.9 |
| | | LaCLIP | 21.5 |
| | | **AlignCLIP** | **23.6** |
| CC12M | ResNet50 | CLIP [†] | 34.6 |
| | | SoftCLIP | 43.2 |
| | | **AlignCLIP** | **46.9** |
| | ViT-B/32 | CLIP[†] | 30.7 |
| | | SoftCLIP | 34.4 |
| | | **AlignCLIP** | **37.8** |
| | ViT-B/16 | CLIP[†] | 36.3 |
| | | SoftCLIP | 42.1 |
| | | LaCLIP | 48.4 |
| | | **AlignCLIP** | **49.1** |
| YFCC15M-V2 | ResNet50 | CLIP [†] | 38.6 |
| | | SoftCLIP | 43.7 |
| | | **AlignCLIP** | **43.8** |
| | ViT-B/32 | CLIP[†] | 32.4 |
| | | SoftCLIP | 35.0 |
| | | **AlignCLIP** | **36.6** |
| | ViT-B/16 | CLIP[†] | 38.9 |
| | | SoftCLIP | 42.4 |
| | | **AlignCLIP** | **44.3** |

[†] Our Implementation

AlignCLIP also has advantages compared with other pre-training methods. Compared with SoftCLIP [9] and LaCLIP [7], AlignCLIP has improved to varying degrees under different settings with less computing resources. In particular, LaCLIP needs to generate multiple augmentation for each caption, while ours only needs to generate visual descriptions parts of them.

In addition, We show the zero-shot classification results of other 6 data sets, where PETS / DTD / F101 / FLOW / SUN / CAL are abbreviations for Pets / Describable Tex- tures / Food-101 / Flowers-102 / SUN397 / Caltech-101 datasets. AVG represents average accuracy. Following previous work [9], we report the results of model pre-trained with YFCC15M-V2. The results are shown in Tab.2. From the table, we can see that AlignCLIP achieves the highest performance in most cases. With Resnet50 as the image encoder, the average accuracy increased from 44.3 in the original CLIP to 50.0, and exceeded SoftCLIP by 0.8 points. Meanwhile, SOTA is also achieved with ViT-B-16 as image encoder.

## 4.5 Linear Probing

We present the results of linear probing in Tab. 3. Experiments were conducted on 7 datasets, namely Pets, Describable Tex-tures,

**Table 2: Zero-shot classification on 6 datasets. The model is pretrained with YFCC15M-V2.**

| Method | Image Encoder | PETS | DTD | F101 | FLOW | SUN | CAL | AVG |
|---|---|---|---|---|---|---|---|---|
| CLIP[†] | | 32.6 | 21.2 | 46.5 | 52.5 | 48.0 | 64.8 | 44.3 |
| SoftCLIP | ResNet50 | 34.9 | **27.1** | 50.8 | **56.3** | 55.9 | 70.4 | 49.2 |
| **AlignCLIP** | | **35.5** | 26.3 | **53.0** | 55.7 | **56.1** | **73.2** | **50.0** |
| CLIP[†] | | 27.2 | 21.6 | 48.3 | 53.8 | 53.4 | 71.5 | 46.0 |
| SoftCLIP | ViT-B/16 | 32.5 | **25.6** | 53.8 | 55.6 | 56.2 | 71.8 | 49.2 |
| **AlignCLIP** | | **35.8** | 24.1 | **54.2** | **56.5** | **56.9** | 72.9 | **50.1** |

[†] Our Implementation

**Table 3: Linear Probing results on 7 datasets.**

| Method | Pretrain Dataset | PETS | DTD | F101 | FLOW | SUN | CAL | ImageNet |
|---|---|---|---|---|---|---|---|---|
| CLIP[†] | | 69.1 | 62.9 | 62.3 | 89.8 | 58.1 | 81.8 | 54.0 |
| LaCLIP | CC3M | 71.1 | 64.0 | 63.8 | 90.2 | **60.2** | 83.3 | 56.5 |
| **AlignCLIP** | | **72.0** | **64.5** | 63.9 | **91.1** | 59.9 | **84.7** | **57.7** |
| CLIP[†] | | 85.9 | 75.1 | 80.9 | 95.1 | 71.5 | 91.5 | 69.9 |
| LaCLIP | CC12M | 87.7 | 75.7 | 82.9 | **96.4** | **73.8** | 93.0 | 72.3 |
| **AlignCLIP** | | **88.9** | **75.9** | **83.8** | 95.5 | 72.6 | **93.9** | **72.9** |

[†] Our Implementation

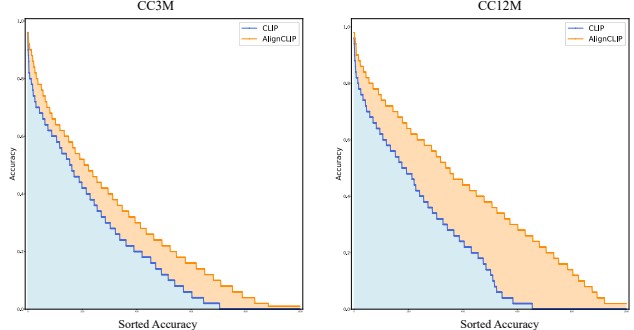

**Figure 3: ImageNet 1K class-wise accuracy for comparison.**

Food-101, Flowers-102, SUN397, Caltech-101 and ImageNet. From the table, we can see that AlignCLIP achieves the highest performance in most situations. On the most convincing ImageNet dataset, AlignCLIP exceeds the CLIP baseline by more than 3 points. At the same time, it also exceeds LaCLIP [7] by about 1 point. It is worth noting that only the image encoder is used in the linear probing setting, while our method is an improvement on the text side, which indicates that improvements on the text encoder can also facilitate the training of image encoders. This is because the improvements on the text input in AlignCLIP have brought richer and more comprehensive supervision signals to the image encoder, thereby improving the performance of the image encoder.

## 4.6 Zero-shot Image-text Retrieval

Here, we verify the effectiveness of our method on image and text retrieval tasks. The experimental results are shown in Tab. 4. All the models are pre-trained on YFCC15M-V2 dataset. As indicated in the table, AlignCLIP achieves the highest performance under identical conditions of pre-training data and image encoder. Particularly, under the ViT-B/16 image encoder, AlignCLIP significantly surpass SoftCLIP and CLIP baseline, which demonstrates that our pre-training method with multi-domain alignment and object-IoU loss can develop a more robust encoder, thereby providing better representations for enhanced retrieval performance.

## 4.7 Ablation Studies

Unless otherwise stated, all experiments are performed with CC3M as pre-training, ResNet50 as image encoder.

*4.7.1 Effects of Different Modules.* Here, we analyze the role and impact of each different module. The specific implementation details are: 1) For models without Object-IoU loss, we use the original CLIP cross-entropy loss to optimize the model. 2) All models have original captions participating in the alignment. 3) For models without visual descriptions, visual descriptions were removed during both training and testing phases. The experimental results are shown in Tab. 5, where the zero-shot classification results of ImageNet 1K is reported. It can be seen from the results that: 1) All modules will bring different levels of improvement. 2) If only one module is added separately, the visual description and object-IoU will bring greater improvements, which improves top1 accuracy by 4.5 and 5.1 points respectively. The test prompt improvement is relatively small, only improves accuracy by about 1 point. 3) The model will all modules achieve highest performance, with 27.0 top1 accuracy.

*4.7.2 Different Implementations of Caption Object Parsing.* In this section, we compare different implementations of caption object parsing, including part-of-speech tagging (POS)and large language models (LLM). We conducted experiments on CC3M and CC12M. In addition to comparing the zero-shot classification performance on imageNet 1K, we also compared the generation time. For LLM, the visual description generation time is obtained under 32 V100s in parallel. For POS, the generation time is obtained in parallel with 64 threads CPU. The experimental results are shown in Tab. 6. It can be seen from the results that the POS solution can be faster, but based on LLM, it can get higher accuracy. Even though the accuracy of POS is relatively lower, it still surpasses SoftCLIP [9] on both CC3M and CC12M, and achieves SOTA.

*4.7.3 Visualization and Qualitative Analysis.* To validate the improvement on tail classes, we visualize the category-by-category accuracy on imagenet 1k trained with our AlignCLIP and CLIP baseline. The results are shown in Fig.3. We conducted experiments on CC3M and CC12M, respectively. As can be seen from Fig.3, AlignCLIP can significantly improve the performance of the tail classes, and the number of categories with an accuracy of 0 is greatly reduced. Meanwhile, due to the introduction of object-IoU loss, the performance of head classes has also been improved.

Furthermore, we also visualized the distribution of image captions, test prompts, and visual descriptions after training with AlignCLIP. The results are shown in Fig.4. As we can see, after training

**Table 4: The results of zero-shot image-text retrieval on Flicker30K and MS-COCO.**

| Method | Image Encoder | Flickr30K(1K) | | | | | | MS-COCO(5K) | | | | | |
| --- | --- | --- | --- | --- | --- | --- | --- | --- | --- | --- | --- | --- | --- |
| | | Image-to-Text | | | Text-to-Image | | | Image-to-Text | | | Text-to-Image | | |
| | | R@1 | R@5 | R@10 | R@1 | R@5 | R@10 | R@1 | R@5 | R@10 | R@1 | R@5 | R@10 |
| CLIP† | ResNet50 | 54.9 | 81.6 | 90.5 | 37.1 | 65.0 | 75.0 | 29.4 | 54.8 | 66.1 | 18.9 | 40.7 | 52.5 |
| DECLIP | | 58.7 | 85.0 | 92.5 | 40.7 | 68.9 | 78.4 | 31.1 | 59.0 | 69.9 | 20.6 | 43.8 | 55.4 |
| SoftCLIP | | 62.1 | 86.4 | **93.0** | 43.0 | 71.0 | **80.3** | 36.0 | 61.2 | 72.3 | 22.2 | 45.8 | 57.3 |
| **AlignCLIP** | | **63.6** | **86.9** | 92.8 | **44.2** | **71.5** | 79.1 | **37.3** | **61.7** | **74.8** | **23.3** | **47.0** | **59.1** |
| CLIP† | ViT-B/16 | 54.9 | 80.0 | 88.4 | 37.2 | 64.3 | 74.3 | 30.7 | 56.2 | 67.4 | 19.1 | 40.9 | 52.5 |
| SoftCLIP | | 56.2 | 82.1 | 88.6 | 37.2 | 64.3 | 74.5 | 30.9 | 56.2 | 68.3 | 19.2 | 41.2 | 52.6 |
| **AlignCLIP** | | **62.1** | **85.8** | **92.0** | **42.6** | **72.5** | **81.9** | **35.7** | **60.6** | **73.4** | **22.2** | **45.6** | **55.0** |

† Our Implementation

**Table 5: Effects of different components. The zero-shot classification accuracy on ImageNet 1K is reported.**

| Visual Des | Test Prompt | Object-IoU | ZS Top1 | ZS Top5 |
| --- | --- | --- | --- | --- |
| | | | 17.8 | 35.9 |
| ✓ | | | 22.3 | 43.5 |
| | ✓ | | 19.1 | 38.7 |
| | | ✓ | 22.9 | 45.6 |
| ✓ | ✓ | | 23.0 | 45.8 |
| ✓ | ✓ | ✓ | **27.0** | **52.2** |

**Table 6: Different implementations of caption object parsing.**

| Methods | CC3M | | CC12M | |
| --- | --- | --- | --- | --- |
| | ZS Top1 | Time(h) | ZS Top1 | Time(h) |
| POS | 24.7 | 1.9 | 44.1 | 7.6 |
| LLM | 27.0 | 13.3 | 46.9 | 52.5 |

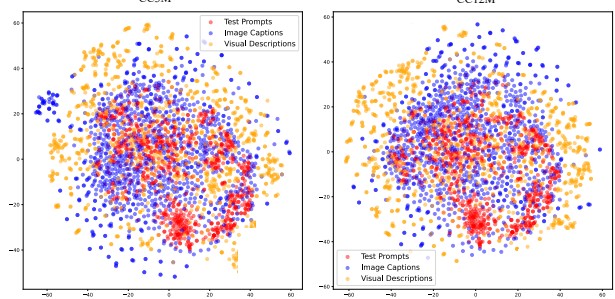

**Figure 4: The t-SNE visualization of feature distribution with AlignCLIP training.**

with AlignCLIP, the distributions of the three have been better aligned, and the distribution shift has been greatly alleviated.

*4.7.4 Effects of Proportions in Visual Descriptions.* Here, we investigate the impact of varying the proportion of visual descriptions in the training and test sets. The experiment was pre-trained on CC3M, and the zero-shot classification results of ImageNet 1K were

**Table 7: Effects of $\alpha$ and $\beta$ in visual descriptions generation.**

| Value of $\alpha$ and $\beta$ | Top 1 | Top5 |
| --- | --- | --- |
| $\alpha = 0, \beta = 0\%$ | 23.1 | 46.3 |
| $\alpha = 0, \beta = 30\%$ | 24.9 | 48.0 |
| $\alpha = 0, \beta = 50\%$ | 25.6 | 49.6 |
| $\alpha = 0, \beta = 100\%$ | 20.2 | 42.7 |
| $\alpha = 10, \beta = 30\%$ | 26.0 | 50.4 |
| $\alpha = 10, \beta = 50\%$ | 26.5 | 51.7 |
| $\alpha = 10, \beta = 70\%$ | 25.9 | 49.8 |
| $\alpha = 30, \beta = 50\%$ | 27.0 | 52.2 |

reported. The results are shown in Tab.7, where $\alpha$ controls the proportion of tail description during training, and $\beta$ controls the proportion of tail during testing. As can be seen from the table: 1) If the model is not aligned using visual descriptions during training, i.e. $\alpha = 0$, the improvement is smaller. 2) Even if 10% of the visual description is added to the training stage, the effect is greatly improved, exceeding the situation without adding visual description. 3) However, it is not necessarily the case that a higher proportion of visual descriptions leads to better performance. We observe a sharp decline in performance when the visual descriptions in the test set reach 100%.

## 5 CONCLUSION

In this paper, we introduce a novel image-text alignment training approach named AlignCLIP. To address the issue of CLIP's lower recognition performance for rare, tail-end classes, we propose utilizing visual descriptions to enhance these tail-end classes. Additionally, we introduce multi-domain alignment to synchronize the distributions of image captions, visual descriptions, and test prompts. To achieve this, we first perform caption object parsing on image captions to identify the object sets they contain, thereby generating samples for alignment. Thanks to the parsed object sets, our approach also implements object-IoU loss at a reduced computational cost, facilitating the computation of soft labels. Our method was pre-trained on multiple image-text datasets and evaluated across various tasks. The results demonstrate that our approach significantly outperforms the CLIP baseline and exceeds existing methods, establishing a new SOTA.

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
