# OpenReview forum: "AlignCLIP: Align Multi Domains of Texts Input for CLIP models with Object-IoU Loss"
_acmmm.org/ACMMM/2024/Conference — MM2024 Poster_

### Official Review · Reviewer_Ygyq · 2024-05-21

**Rating:** 1
**Confidence:** 4

**Summary:**

The paper propose a method to overcome the low performance of CLIP on longtailed concepts. They use existing techniques of generating visual descriptions for rare classes using LLMs. Then they hypothesise that the train, test, and visual description captions have a distribution shift which does not allow to effectively learn the new descriptions. The authors propose to align the three caption types to improve the performance on longtailed classes.

**Strengths:**

Performance improvement in the setting of small scale pretraining.

**Limitations:**

1) The class wise performance analysis is done on CLIP trained from scratch by the authors on CC3M dataset and not on the original CLIP, which does not represent the statistics of the original model. How does the statistics hold true for the original CLIP?

2) The entire paper is built upon the hypothesis of distribution shift, performed on small scale CLIP trained from scratch on the CC3M. While, CLIP is trained on 400M internet captions, how does it translate on large scale CLIIP.

3) The evaluation and comparison is also done on baseline CLIP trained on CC3M which again might not translate on large scale.

4) How do you claim SOTA when you don't compare at large sale CLIP?

**Suitability:**

2

---

### Official Review · Reviewer_aUfn · 2024-05-22

**Rating:** 5
**Confidence:** 3

**Summary:**

In this paper, the authors introduce a novel image-text alignment training approach named AlignCLIP. To address the issue of CLIP’s
lower recognition performance for rare, tail-end classes, they propose utilizing visual descriptions to enhance these tail-end classes.
Additionally, they introduce multi-domain alignment to synchronize the distributions of image captions, visual descriptions, and test
prompts. To achieve this, they first perform caption object parsing on image captions to identify the object sets they contain, thereby
generating samples for alignment. Thanks to the parsed object sets, proposed approach also implements object-IoU loss at a reduced
computational cost, facilitating the computation of soft labels. The proposed  method was pre-trained on multiple image-text datasets and evaluated across various tasks. The results demonstrate that proposed approach significantly outperforms the CLIP baseline and exceeds existing methods, establishing a new SOTA.

**Strengths:**

a) The experiment in this paper is thorough, and the presentation is relatively clear.

b) The proposed  method significantly surpasses the CLIP baseline and exceeds existing methods, achieving a new SOTA.

c) The motivation of the paper is reasonable, concise, and effective.

**Limitations:**

a)  The rightmost part of Figure 2 is unclear and not intuitive, it needs to be revised.

b) The comparative methods in the table need to be accompanied by reference numbers for easy reading.

**Suitability:**

3

---

### Official Review · Reviewer_aHoh · 2024-05-24

**Rating:** 4
**Confidence:** 4

**Summary:**

The paper introduces AlignCLIP, a novel approach to improve the performance of CLIP  models in recognizing rare and tail-end classes, which are often challenging due to their infrequent occurrence in training datasets. The authors propose a method that involves aligning multiple domains of text input, namely image captions, visual descriptions, and test prompt templates, to address the distribution shifts that occur between these domains.

**Strengths:**

1. The paper introduces a novel approach to address the long-tail recognition problem in vision-language models, which is a significant challenge in the field. The concept of aligning multiple text domains (image captions, visual descriptions, and test prompts) through caption object parsing and the introduction of Object-IoU Loss is innovative.
2. The theoretical foundation of AlignCLIP is sound, as it builds upon the established framework of CLIP models while innovating to address specific limitations. The use of soft labels and many-to-many relationships between images and texts is a theoretically justified advancement over traditional one-to-one matching approaches.
3. The paper is well-structured, with clear explanations of the methodology, experiments, and results. Figures and tables are used effectively to visualize the performance improvements and distribution alignments, which aids in understanding the impact of the proposed approach.

**Limitations:**

1. The evaluation is conducted primarily on existing benchmark datasets. There might be a need for further evaluation on more diverse or newer datasets to ensure the robustness of the approach. I'd like to take a look at the comparison on a dataset that contains a large number of rare classes, such as Things(https://osf.io/jum2f/).
2. The caption object parsing relies on part-of-speech tagging and large language models. The performance of AlignCLIP could be influenced by the quality and accuracy of these external tools, which may not always be consistent.
3. While the paper claims reduced computational cost due to the Object-IoU Loss, the overall training and inference process might still be resource-intensive, especially when using large language models for parsing and generating descriptions.
4. While the paper compares AlignCLIP with the original CLIP and some variants, it would be beneficial to see comparisons (such as FLIP) with a broader range of contemporary methods to better understand its relative performance.

**Suitability:**

3

---

### Official Review · Reviewer_fzks · 2024-05-25

**Rating:** 5
**Confidence:** 2

**Summary:**

The paper introduces AlignCLIP, a novel approach for enhancing CLIP models through multi-domain alignment and Object-IoU Loss. By leveraging caption object parsing, AlignCLIP generates visual descriptions and test prompts to improve recognition of rare classes. The method aligns visual descriptions, image captions, and test prompts to create category prototypes, leading to enhanced classification accuracy. AlignCLIP surpasses existing methods with lower computational costs, making it a promising advancement in vision-language pre-training.

**Strengths:**

1, AlignCLIP introduces a unique method for addressing the long-tail distribution issue in CLIP models by aligning visual descriptions, image captions, and test prompts. This innovative multi-domain alignment approach sets it apart from traditional methods . The incorporation of object-IoU loss for generating soft labels based on the similarity between object sets in a batch is a novel contribution to the field.

2, The paper is well-written, presenting the methodology and results in a clear and concise manner, making it accessible to a wide audience . AlignCLIP's application in vision-language pre-training opens up possibilities for improved performance in downstream tasks by addressing the long-tail problem and enhancing alignment between visual and textual modalities.

3, The use of object parsing from image captions for multi-domain alignment is technically robust and offers a practical solution to the distribution shift problem during training . The inclusion of sufficient experimental evaluations across various tasks showcases the effectiveness and robustness of the AlignCLIP model.

**Limitations:**

1,  The paper lacks a comprehensive and detailed explanation of the pre-training process in AlignCLIP. Insufficient elaboration on the specific pre-training methodology, model architecture, hyperparameter selection, and training procedures may hinder reproducibility and understanding of the pre-training phase. This lack of clarity makes it challenging to assess the benefits brought about by the pre-training process in AlignCLIP accurately.

2, Future experiments could be expanded to include a broader range of datasets beyond COCO and Flickr30K for more comprehensive evaluations.

3, While AlignCLIP builds upon the success of CLIP in vision-language alignment, it is known that CLIP may lack certain reasoning capabilities. Have experiments been conducted using other large language models (LLMs) in Align Multi Domains of Texts Input? Since LLMs are known for their enhanced reasoning abilities, could leveraging these models lead to improved alignment performance?

**Suitability:**

2

---

### Meta-Review · Area_Chair_fwTT · 2024-06-30

**Recommendation:** Accept (Poster)
**Confidence:** 5

**Metareview:**

The paper initially received three positive indications (2WA, BA) and one negative (R) indication. After the rebuttal, the ratings have remained almost unchanged, with positive reviewers keeping a positive indication (2WA, BA), and the negative reviewer raising his/her score while still maintaining a negative indication (WR). Most reviewers have appreciated the novelty of the approach and the soundness of the experimental results. Concerns raised by the negative reviewer were related to the absence of comparisons with the original OpenAI CLIP model. The AC, however, agrees with the authors and the other three reviewers that it is standard practice to re-train CLIP on publicly available datasets. The paper can be accepted as poster.